# An Elastodiffusive Orthotropic Euler–Bernoulli Beam Considering Diffusion Flux Relaxation

**Dmitry Tarlakovskii [1,2] and Andrei Zemskov [1,2,*]**

[1]   Research Institute of Mechanics, Lomonosov Moscow State University, 119192 Moscow, Russia;
     tdvhome@mail.ru
[2]   Moscow Aviation Institute (National Research University), 125993 Moscow, Russia
*   Correspondence: azemskov1975@mail.ru; Tel.: +7-926-522-38-24

**Abstract:** This article considers an unsteady elastic diffusion model of Euler–Bernoulli beam oscillations in the presence of diffusion flux relaxation. We used the model of coupled elastic diffusion for a homogeneous orthotropic multicomponent continuum to formulate the problem. A model of unsteady bending for the elastic diffusive Euler–Bernoulli beam was obtained using Hamilton's variational principle. The Laplace transform on time and the Fourier series expansion by the spatial coordinate were used to solve the obtained problem.

**Keywords:** elastic diffusion; coupled problem; unsteady problem; Green's function; integral transformation; multicomponent continuum; Euler–Bernoulli beam

---

## 1. Introduction

In order to solve various types of fundamental, applied, and technical problems, scientists and engineers often have to consider the unsteady interaction between various physical fields. An example of this is the interaction between mechanical and diffusion fields. Among the recent publications, it can be noted that [1–19] were devoted to this problem. In particular, [1–11] considered the thermal effects, while the electromagnetic effects were studied in [12–15]. In addition, the analysis of fast unsteady processes in relatively short time periods requires the relaxation of thermal and diffusion perturbations to be taken into account [1,2,4–6,8,9,11–13].

Based on the reviewed publications, it can be concluded that numerical–analytical methods constructed on the Laplace and Fourier integral transformations are used to solve unsteady mechanodiffusion problems. In this case, the Durbin method is mainly used for the Laplace transform inversion. This method allows the Mellin integral to be expressed through the Fourier transform. Special numerical algorithms are used to do the inverse Fourier transform. Descriptions of these algorithms can be found in [20–22].

However, there is no universal algorithm for the inversion of Laplace transform, as noted in [23]. Each specific algorithm fits a certain class of functions. At the same time, the specificity of Laplace images influences the choice of suitable systems of functions with which one can approximate the unknown original values. In addition, Laplace images are very cumbersome to obtain by solving the coupled problems. It is not always possible in practice to verify the capability of using one or the other algorithm to find Laplace transform originals. Thus, it can be concluded that the Laplace transform inversion is the main mathematical complexity that arises when solving unsteady problems (mechanodiffusion problems in particular).

This article considers an unsteady elastic diffusion problem of Euler–Bernoulli beam oscillations in the presence of diffusion flux relaxation. It also proposes a method to construct a solution based on the use of the integral Laplace transform and expansion into series by eigenfunctions.

The expansion coefficients are represented as rational functions, which simplifies the issue of inverse Laplace transform. Thus, the problem considered in the article is solved analytically.

It should be noted that analogues of the considered problem are absent in well-known scientific publications.

## 2. Problem Formulation

We consider the unsteady oscillation problem of elastic diffusion Euler–Bernoulli beam in the presence of diffusion flux relaxation. The beam in general formulation is under the action of longitudinal and transverse forces. Also, bending moments are at its ends. The load scheme is presented in Figure 1.

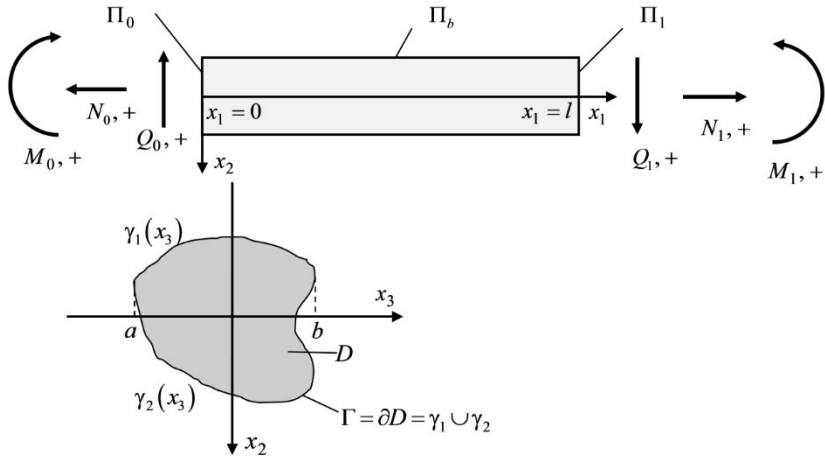

**Figure 1.** The formulation of the problem.

For the mathematical problem formulation, we use the coupled elastic diffusion continuum model in a rectangular Cartesian coordinate system, which has the following form [1–18]:

$$\ddot{u}_i = \frac{\partial \sigma_{ij}}{\partial x_j} + F_i, \quad \dot{\eta}^{(q)} = -\frac{\partial J_i^{(q)}}{\partial x_i} + Y^{(q)} \quad \left(q = \overline{1, N}\right). \tag{1}$$

where the point denotes a time derivative; $\sigma_{ij}$ and $J_i^{(q)}$ are the stress tensor components and the diffusion flux vector, respectively. They are defined as follows (the beam material is perfect solid solution):

$$\sigma_{ij} = C_{ijkl} \frac{\partial u_k}{\partial x_l} - \sum_{q=1}^{N} \alpha_{ij}^{(q)} \eta^{(q)}, \quad J_i^{(q)} + \tau^{(q)} \dot{J}_i^{(q)} = -D_{ij}^{(q)} \frac{\partial \eta^{(q)}}{\partial x_j} + \Lambda_{ijkl}^{(q)} \frac{\partial^2 u_k}{\partial x_j \partial x_l} \quad \left(q = \overline{1, N}\right). \tag{2}$$

Here, the dots denote the time derivative. All quantities in (1) and (2) are dimensionless. For them, the following notations are used:

$$x_i = \frac{x_i^*}{l}, \quad u_i = \frac{u_i^*}{l}, \quad \tau = \frac{Ct}{l}, \quad C_{ijkl} = \frac{C_{ijkl}^*}{C_{1111}}, \quad C^2 = \frac{C_{1111}^*}{\rho}, \quad \alpha_{ij}^{(q)} = \frac{\alpha_{ij}^{*(q)}}{C_{1111}},$$

$$\tau_q = \frac{C\tau^{(q)}}{L}, \quad D_{ij}^{(q)} = \frac{D_{ij}^{*(q)}}{Cl}, \quad \Lambda_{ijkl}^{(q)} = \frac{m^{(q)} D_{ij}^{*(q)} \alpha_{kl}^{*(q)} n_0^{(q)}}{\rho R T_0 C l}, \quad F_i = \frac{\rho l F_i^*}{S_{1111}}, \quad Y^{(q)} = \frac{l Y^{*(q)}}{C}, \tag{3}$$

where $t$ is time; $x_i^*$ are rectangular Cartesian coordinates; $\rho$ is the medium density; $u_i^*$ are displacement vector components; $C_{ijkl}^*$ are elastic constant tensor components; $T_0$ is the initial temperature; $D_{ij}^{*(q)}$ are the self-diffusion coefficients; $R$ is the universal gas constant; $m^{(q)}$ is the molar mass; $\eta^{(q)} = n^{(q)} - n_0^{(q)}$

is the concentration increment of $q$-th component in the $N$-component medium; $n_0^{(q)}$ and $n^{(q)}$ are the initial and actual concentrations (mass fractions); $\alpha_{ij}^{*(q)}$ are coefficients characterizing the medium volumetric changes due to diffusion; $l$ is beam length; $F_i$ and $Y^{(q)}$ are mechanical and diffusive bulk perturbations; and $\tau^{(q)}$ is the relaxation time of diffusion perturbations.

The formulation of the problem is completed by the initial and boundary conditions.

*Initial conditions*:

$$u_i|_{\tau=t_0} = u_{i0}, \quad \dot{u}_i|_{\tau=t_0} = v_{i0}, \quad \eta^{(q)}\Big|_{\tau=t_0} = \eta_0^{(q)} \quad (q = \overline{1,N}). \tag{4}$$

Here, $u_{i0}, v_{i0}, \eta_0^{(q)}$ are the given functions of spatial coordinates. Further in the paper, we assume that $t_0 = 0, \quad u_{i0} = 0, \quad v_{i0} = 0, \quad \eta_0^{(q)} = 0$.

*Boundary conditions* (domain $G$ is bounded; $n_i$ are components of the outer normal unit vector to $\partial G$, $\partial G = \Pi_u \cup \Pi_\sigma = \Pi_\eta \cup \Pi_J$):

$$u_i|_{\Pi_u} = U_i, \quad \sigma_{ij}n_j\big|_{\Pi_\sigma} = P_i, \quad \eta^{(q)}\Big|_{\Pi_\eta} = N^{(q)}, \quad \left(J_i^{(q)} + \tau_q \dot{J}_i^{(q)}\right)\Big|_{\Pi_J} = I_i^{(q)} \quad (\tau > 0, q = \overline{1,N}). \tag{5}$$

The quantities on the boundary conditions on right sides are surface kinematic $U_i$, $N^{(q)}$ and dynamic $P_i$, $I_i^{(q)}$ perturbations.

To construct the beam bending equations, a transition to the variational formulation of the problem (1)–(5) is used. According to Hamilton's variational principle, the relations (1)–(5) can be regarded as a condition for the stationarity of a certain functional $H\left(u_i, \eta^{(q)}\right)$, whose variation is written thus:

$$\delta H = \int_{t_1}^{t_2} d\tau \int_G \left(\ddot{u}_i - \frac{\partial \sigma_{ij}}{\partial x_j} - F_i\right)\delta u_i dG + \sum_{q=1}^{N} \int_{t_1}^{t_2} d\tau \int_G \left(1 + \tau_q \frac{\partial}{\partial \tau}\right)\left(\eta^{(q)} + \frac{\partial J_i^{(q)}}{\partial x_i} - Y^{(q)}\right)\delta\eta^{(q)} dG +$$

$$+ \int_{t_1}^{t_2} \iint_{\Pi_\sigma} (\sigma_{ij}n_j - P_i)\delta u_i dS d\tau + \sum_{q=1}^{N} \int_{t_1}^{t_2} \iint_{\Pi_J} \left(J_i^{(q)} + \tau_q \dot{J}_i^{(q)} - I_i^{(q)}\right)n_i \delta\eta^{(q)} dS d\tau. \tag{6}$$

Further, we will assume that

i　　The axis $Ox_3$ is the central axis of the cross section. In this case,

$$\iint_D x_2 dx_2 dx_3 = 0. \tag{7}$$

ii　　The side surface $\Pi_b$ is free from mechanical loads, i.e.,

$$\sigma_{ij}n_j\big|_{\Pi_b} = 0. \tag{8}$$

iii　　We also assume that there is no mass transfer through the side surface,

$$\left(J_i^{(q)} + \tau_q \dot{J}_i^{(q)}\right)\Big|_{\Pi_b} = 0. \tag{9}$$

iv　　The beam material is a homogeneous ortotropic continuum.

The bending of beam is considered in plane $x_1 O x_2$. Then $u_k = u_k(x_1, x_2, \tau)$, $k = 1, 2$; $u_3 = 0$; $\varepsilon_{i3} = 0$. Mass transfer occurs also in the plane $x_1 O x_2$, i.e., $\eta^{(q)} = \eta^{(q)}(x_1, x_2, \tau)$.

v    Transverse deflections are considered small. Then, the linearization of the unknown quantities with respect to the variable $x_2$ will have the following form:

$$u_1(x_1, x_2, \tau) = u(x_1, \tau) - x_2\chi(x_1, \tau), \quad u_2(x_1, x_2, \tau) = v(x_1, \tau) + x_2\psi(x_1, \tau),$$

$$\eta^{(q)}(x_1, x_2, \tau) = N_q(x_1, \tau) + x_2 H_q(x_1, \tau). \tag{10}$$

vi   The cross sections after deformation remain normal to the neutral line of the beam (Euler–Bernoulli 's beam theory). Also, we assume that there are no deformations along the $Ox_2$ axis [24,25].

$$\varepsilon_{22} = \frac{\partial u_2}{\partial x_2} = \psi = 0 \Rightarrow \psi = 0, \quad \chi(x_1, \tau) = v'(x_1, \tau).$$

Then, Equation (10) will take the following form:

$$u_1 = u - x_2 v', \quad u_2 = v, \quad u_3 = 0, \quad u = u(x_1, \tau), \quad v = v(x_1, \tau). \tag{11}$$

Here, the prime denotes the derivative with respect to the variable $x_1$.
From (8) and (9), it follows that,

$$\sigma_{22} n_2|_{\Pi_b} = \sigma_{22} n_2|_{\Gamma} = \sigma_{22}|_{\gamma_2(x_3)}^{\gamma_1(x_3)} = 0; \tag{12}$$

$$\int_G \frac{\partial \sigma_{22}}{\partial x_2} dG = \int_0^1 dx_1 \iint_D \frac{\partial \sigma_{22}}{\partial x_2} dx_2 dx_3 = \int_0^1 dx_1 \int_a^b \int_{\gamma_2(x_3)}^{\gamma_1(x_3)} \frac{\partial \sigma_{22}}{\partial x_2} dx_2 dx_3 = \int_0^1 dx_1 \int_a^b \sigma_{22}|_{\gamma_2(x_3)}^{\gamma_1(x_3)} dx_3 = 0. \tag{13}$$

The components of the stress tensor and the diffusion flow vector will have the following form:

$$\sigma_{11} = C_{11}\frac{\partial u_1}{\partial x_1} + C_{12}\frac{\partial u_2}{\partial x_2} - \sum_{q=1}^N \alpha_1^{(q)}\eta^{(q)} = (u' - x_2 v'') - \sum_{q=1}^N \alpha_1^{(q)}(N_q + x_2 H_q), \quad \sigma_{12} = C_{66}\left(\frac{\partial u_1}{\partial x_2} + \frac{\partial u_2}{\partial x_1}\right) = 0,$$

$$\sigma_{22} = C_{12}\frac{\partial u_1}{\partial x_1} + C_{22}\frac{\partial u_2}{\partial x_2} - \sum_{q=1}^N \alpha_2^{(q)}\eta^{(q)} = C_{12}(u' - x_2 v'') - \sum_{q=1}^N \alpha_2^{(q)}(N_q + x_2 H_q),$$

$$J_1^{(q)} + \tau_q \dot{J}_1^{(q)} = -D_1^{(q)}\frac{\partial \eta^{(q)}}{\partial x_1} + \Lambda_{11}^{(q)}\frac{\partial^2 u_1}{\partial x_1^2} + \Lambda_{12}^{(q)}\frac{\partial^2 u_2}{\partial x_1 \partial x_2} = -D_1^{(q)}\left(N_q' + x_2 H_q'\right) + \Lambda_{11}^{(q)}(u'' - x_2 v'''), \tag{14}$$

$$J_2^{(q)} + \tau_q \dot{J}_2^{(q)} = -D_2^{(q)}\frac{\partial \eta^{(q)}}{\partial x_2} + \Lambda_{22}^{(q)}\frac{\partial^2 u_2}{\partial x_2^2} + \Lambda_{21}^{(q)}\frac{\partial^2 u_1}{\partial x_1 \partial x_2} = -D_2^{(q)} H_q - \Lambda_{21}^{(q)} v'' \quad (q = \overline{1, N}),$$

where

$$C_{11} = C_{1111}, \quad C_{12} = C_{1122}, \quad C_{22} = C_{2222}, \quad C_{66} = C_{1212}, \quad \alpha_1^{(q)} = \alpha_{11}^{(q)}, \quad \alpha_2^{(q)} = \alpha_{22}^{(q)},$$

$$D_1^{(q)} = D_{11}^{(q)}, \quad D_2^{(q)} = D_{22}^{(q)}, \quad \Lambda_{11}^{(q)} = \Lambda_{1111}^{(q)}, \quad \Lambda_{12}^{(q)} = \Lambda_{1122}^{(q)}, \quad \Lambda_{21}^{(q)} = \Lambda_{2211}^{(q)}, \quad \Lambda_{22}^{(q)} = \Lambda_{2222}^{(q)}.$$

Substituting the relations (7), (10)–(14) into (6), we obtain the following:

$$\delta H = \int_{t_1}^{t_2} d\tau \int_0^1 \left[ F\left( \ddot{u} - u'' + \sum_{q=1}^N \alpha_1^{(q)} N' \right) - n \right] \delta u \, dx_1 - \int_{t_1}^{t_2} d\tau \int_0^1 \left[ J_3\left( \ddot{v}'' - v^{IV} - \sum_{q=1}^N \alpha_1^{(q)} H_q'' \right) + m' \right] \delta v \, dx_1 +$$

$$+ \int_{t_1}^{t_2} d\tau \int_0^1 (F\ddot{v} - q)\delta v \, dx_1 + \sum_{q=1}^N \int_{t_1}^{t_2} d\tau \int_0^1 \left[ F\left( N_q + \tau^{(q)} \ddot{N}_q - D_1^{(q)} N_q'' + \Lambda_{11}^{(q)} u''' \right) - y^{(q)} \right] \delta N_q \, dx_1 +$$

$$+ \sum_{q=1}^N \int_{t_1}^{t_2} d\tau \int_0^1 \left[ J_3\left( H_q + \tau^{(q)} \ddot{H}_q - D_1^{(q)} H_q'' - \Lambda_{11}^{(q)} v^{IV} \right) - z^{(q)} \right] \delta H_q \, dx_1 +$$

$$+ \int_{t_1}^{t_2} \left[ J_3\left( \ddot{v}' - v''' - \sum_{q=1}^N \alpha_1^{(q)} H_q' \right) + m \right] \delta v \Bigg|_{x_1=0}^{x_1=1} d\tau +$$

$$+ \int_{t_1}^{t_2} \left\{ \left[ F\left( u' - \sum_{q=1}^N \alpha_1^{(q)} N_q \right) - N \right] \delta u + \left[ J_3\left( v'' + \sum_{q=1}^N \alpha_1^{(q)} H_q \right) + M \right] \delta v' - Q\delta v \right\} \Bigg|_{x_1=0}^{x_1=1} d\tau +$$

$$+ \sum_{q=1}^N \int_{t_1}^{t_2} \left\{ \left[ F\left( -D_1^{(q)} N_q' + \Lambda_{11}^{(q)} u'' \right) - \Gamma^{(q)} \right] \delta N_q - \left[ J_3\left( D_1^{(q)} H_q' + \Lambda_{11}^{(q)} v''' \right) + \Omega^{(q)} \right] \delta H_q \right\} \Bigg|_{x_1=0}^{x_1=1} d\tau.$$

We introduce the following notation:

- $\iint\limits_D dx_2 dx_3 = F$ is the cross-sectional area,

- $\iint\limits_D x_2^2 dx_2 dx_3 = J_3$ is moment of inertia of the beam section relative to the axis $Ox_3$,

- $\iint\limits_D F_1 dx_2 dx_3 = n$ is the linearly distributed axial load,

- $\iint\limits_D F_1 x_2 dx_2 dx_3 = m$ is the linearly distributed moment,

- $\iint\limits_D F_2 dx_2 dx_3 = q$ is the linearly distributed transverse load,

- $\iint\limits_D \left( Y^{(q)} + \tau^{(q)} Y^{(q)} \right) dx_2 dx_3 = y^{(q)}$ is the linear density of bulk mass transfer sources,

$$\iint\limits_D \left( Y^{(q)} + \tau^{(q)} Y^{(q)} \right) x_2 dx_2 dx_3 = z^{(q)},$$

$$N(x_1) = \iint\limits_D P_1(x_1, x_2, x_3, \tau) dx_2 dx_3, \quad M(x_1) = \iint\limits_D P_1(x_1, x_2, x_3, \tau) x_2 dx_2 dx_3, \quad Q(x_1) = \iint\limits_D P_2(x_1, x_2, x_3, \tau) dx_2 dx_3,$$

$$\Gamma^{(q)}(x_1) = \iint\limits_D I_1^{(q)}(x_1, x_2, x_3, \tau) dx_2 dx_3, \quad \Omega^{(q)}(x_1) = \iint\limits_D x_2 I_1^{(q)}(x_1, x_2, x_3, \tau) dx_2 dx_3.$$

Thus, the following boundary-value problems are the necessary conditions for the stationarity of the Hamilton functional considering (7)–(11):

$$\ddot{u} = u'' - \sum_{q=1}^N \alpha_1^{(q)} N_q' + \frac{n}{F}, \quad \dot{N}_q + \tau_q \ddot{N}_q = D_1^{(q)} N_q'' - \Lambda_{11}^{(q)} u''' + \frac{y^{(q)}}{F}; \tag{15}$$

$$\left( u' - \sum_{q=1}^N \alpha_1^{(q)} N_q \right) \Bigg|_{x_1=0} = \frac{N_0}{F}, \quad \left( u' - \sum_{q=1}^N \alpha_1^{(q)} N_q \right) \Bigg|_{x_1=1} = \frac{N_1}{F}; \tag{16}$$

$$\left(-D_1^{(q)} N_q' + \Lambda_{11}^{(q)} u''\right)\Big|_{x_1=0} = \frac{\Gamma_0^{(q)}}{F}, \quad \left(-D_1^{(q)} N_q' + \Lambda_{11}^{(q)} u''\right)\Big|_{x_1=1} = \frac{\Gamma_1^{(q)}}{F}; \tag{17}$$

$$\ddot{v}'' - \frac{F}{J_3} \ddot{v} = v^{IV} + \sum_{j=1}^{N} \alpha_1^{(q)} H_j'' - \frac{q}{J_3} - \frac{m'}{J_3},$$
$$\dot{H}_q + \tau_q \ddot{H}_q = D_1^{(q)} H_q'' + \Lambda_{11}^{(q)} v^{IV} + \frac{z^{(q)}}{J_3}; \tag{18}$$

$$\left(v'' + \sum_{j=1}^{N} \alpha_1^{(q)} H_j\right)\Big|_{x_1=0} = -\frac{M_0}{J_3}, \quad \left(v'' + \sum_{j=1}^{N} \alpha_1^{(q)} H_j\right)\Big|_{x_1=1} = -\frac{M_1}{J_3}; \tag{19}$$

$$\left(v''' + \sum_{j=1}^{N} \alpha_1^{(q)} H'_j - \ddot{v}'\right)\Big|_{x_1=0} = \frac{Q_0 + m|_{x_1=0}}{J_3}, \quad \left(v''' + \sum_{j=1}^{N} \alpha_1^{(q)} H'_j - \ddot{v}'\right)\Big|_{x_1=1} = \frac{Q_1 + m|_{x_1=1}}{J_3}; \tag{20}$$

$$\left(D_1^{(q)} H_q' + \Lambda_{11}^{(q)} v'''\right)\Big|_{x_1=0} = -\frac{\Omega_0^{(q)}}{J_3}, \quad \left(D_1^{(q)} H_q' + \Lambda_{11}^{(q)} v'''\right)\Big|_{x_1=1} = -\frac{\Omega_1^{(q)}}{J_3}, \tag{21}$$

where

$$N_0 = N(0), \quad N_1 = N(1), \quad M_0 = M(0), \quad M_1 = M(1), \quad Q_0 = Q(0), \quad Q_1 = Q(1),$$
$$\Gamma_0^{(q)} = \Gamma^{(q)}(0), \quad \Gamma_1^{(q)} = \Gamma^{(q)}(1), \quad \Omega_0^{(q)} = \Omega^{(q)}(0), \quad \Omega_1^{(q)} = \Omega^{(q)}(1).$$

In accordance with the variational principle of Lagrange, the boundary conditions (15)–(21) combined with the kinematic boundary conditions are as follows:

$$u|_{x_1=0} = U_0, \quad u|_{x_1=1} = U_1; \tag{22}$$

$$N_q|_{x_1=0} = N_{q0}, \quad N_q|_{x_1=1} = N_{q1}; \tag{23}$$

$$v|_{x_1=0} = V_0, \quad v|_{x_1=1} = V_1; \tag{24}$$

$$v'|_{x_1=0} = V_0', \quad v'|_{x_1=1} = V_1'; \tag{25}$$

$$H_q|_{x_1=0} = H_{q0}, \quad H_q|_{x_1=1} = H_{q1}; \tag{26}$$

$$H_q'|_{x_1=0} = H_{q0}', \quad H_q'|_{x_1=1} = H_{q1}'. \tag{27}$$

Problems (15), (16), and (23) as well as (15), (17), and (22) are equivalent to one-dimensional problems of elastic diffusion for a layer. Their solutions are constructed using Laplace transform and decomposition into trigonometric Fourier series [26–28]. The papers [29,30] also considered problems with other combinations of boundary conditions that are solved by the method of equivalent boundary conditions.

Thus, the object of further consideration in this article will be initial boundary value problems for Equation (18) with a combination of boundary conditions (19), (20), and (24)–(27). Initial conditions will be assumed to be zero.

### 3. Method of Solution

We consider the problems (18), (19), (24), (26). The solutions of the problem are represented in the form ($q = \overline{1, N+1}$):

$$v(x,\tau) = \sum_{k=1}^{N+2} \int_0^\tau [G_{1k}(x, \tau-t)f_{k1}(t) + G_{1k}(1-x, \tau-t)f_{k2}(t)]dt + \sum_{k=1}^{N+1} \int_0^\tau \int_0^1 \widetilde{G}_{1k}(x, \xi, \tau-t)F_k(\xi, t)d\xi dt,$$

$$H_q(x,\tau) = \sum_{k=1}^{N+2} \int_0^\tau \left[ G_{q+1,k}(x, \tau-t)f_{k1}(t) + G_{q+1,k}(1-x, \tau-t)f_{k2}(t) \right]dt +$$

$$+ \sum_{k=1}^{N+1} \int_0^\tau \int_0^1 \widetilde{G}_{q+1,k}(x, \xi, \tau-t)F_k(\xi, t)d\xi dt.$$

(28)

Here, $x = x_1$; $G_{mk}$ and $\widetilde{G}_{mk}$ are surface and bulk Green's functions; $F_1(x, \tau) = (q + m')/J_3$ and $F_{q+1}(x, \tau) = z^{(q)}/J_3$ are body forces entering into Equation (18); and $f_{kl}(t)$ are surface disturbances that have the following form:

$$f_{11}(\tau) = -\frac{M_0(\tau)}{J_3}, \quad f_{12}(\tau) = -\frac{M_1(\tau)}{J_3},$$

$$f_{21}(\tau) = V_0, \quad f_{22}(\tau) = V_1, \quad f_{q+2,1}(\tau) = H_{q0}(\tau), \quad f_{q+2,2}(\tau) = H_{q1}(\tau).$$

The surface Green's functions $G_{mk}$ satisfy

$$\ddot{G}_{1k}'' - \frac{F}{J_3}\ddot{G}_{1k} = G_{1k}^{IV} + \sum_{j=1}^N \alpha_1^{(j)} G_{j+1,k}'',$$

$$\dot{G}_{q+1,k} + \tau_q \ddot{G}_{q+1,k} = D_1^{(q)} G_{q+1,k}'' + \Lambda_{11}^{(q)} G_{1k}^{IV},$$

(29)

$$\left. \left( G_{1k}'' + \sum_{j=1}^N \alpha_1^{(j)} G_{j+1,k} \right) \right|_{x_1=0} = \delta_{1k}\delta(\tau), \quad \left. \left( G_{1k}'' + \sum_{j=1}^N \alpha_1^{(j)} G_{j+1,k} \right) \right|_{x_1=1} = 0,$$

$$\left. G_{1k} \right|_{x_1=0} = \delta_{2k}\delta(\tau), \quad \left. G_{1k} \right|_{x_1=1} = 0, \quad \left. G_{q+1,k} \right|_{x_1=0} = \delta_{q+2,k}\delta(\tau), \quad \left. G_{q+1,k} \right|_{x_1=1} = 0.$$

(30)

The bulk Green's functions $\widetilde{G}_{mk}$ satisfy

$$\ddot{\widetilde{G}}_{1k}'' - \frac{F}{J_3}\ddot{\widetilde{G}}_{1k} = \widetilde{G}_{1k}^{IV} + \sum_{j=1}^N \alpha_1^{(j)} \widetilde{G}_{j+1,k}'' - \delta_{1k}\delta(x-\xi)\delta(\tau),$$

$$\dot{\widetilde{G}}_{q+1,k} + \tau_q \ddot{\widetilde{G}}_{q+1,k} = D_1^{(q)} \widetilde{G}_{q+1,k}'' + \Lambda_{11}^{(q)} \widetilde{G}_{1k}^{IV} + \delta_{q+1,k}\delta(x-\xi)\delta(\tau),$$

and homogeneous boundary conditions corresponding to (30).

We consider the problem of finding the surface Green's functions $G_{mk}$. Applying the Laplace transformation with respect to time to (29) and (30), we get the following ($s$ is the Laplace transform parameter):

$$s^2 G_{1k}^{L''}(x,s) - \frac{F}{J_3} s^2 G_{1k}^L(x,s) = \left[G_{1k}^{IV}(x,s)\right]^L + \sum_{j=1}^{N} \alpha_1^{(j)} G_{j+1,k}^{L''}(x,s),$$

$$\left(s + \tau_q s^2\right) G_{q+1,k}^L(x,s) = D_1^{(q)} G_{q+1,k}^{L''}(x,s) + \Lambda_{11}^{(q)} \left[G_{1k}^{IV}(x,s)\right]^L,$$

$$\left(G_{1k}^{L''} + \sum_{j=1}^{N} \alpha_j G_{j+1,k}^L\right)\Bigg|_{x_1=0} = \delta_{1k}, \quad \left(G_{1k}^{L''} + \sum_{j=1}^{N} \alpha_j G_{j+1,k}^L\right)\Bigg|_{x_1=1} = 0,$$

$$G_{1k}^L\Big|_{x_1=0} = \delta_{2k}, \quad G_{1k}^L\Big|_{x_1=1} = 0, \quad G_{q+1,k}^L\Big|_{x_1=0} = \delta_{q+1,k}, \quad G_{q+1,k}^L\Big|_{x_1=1} = 0.$$

Next, we multiply each equation by $\sin \lambda_n x$ and integrate in the interval $[0, 1]$. The result is as follows:

$$k_1(\lambda_n, s) G_{1k}^{Ls}(\lambda_n, s) - \lambda_n^2 \sum_{j=1}^{N} \alpha_1^{(j)} G_{j+1,k}^{Ls}(\lambda_n, s) = F_{1k}(\lambda_n),$$

$$-\lambda_n^4 \Lambda_{11}^{(q)} G_{1k}^{Ls}(\lambda_n, s) + k_{q+1}(\lambda_n, s) G_{q+1,k}^{Ls}(\lambda_n, s) = F_{q+1,k}(\lambda_n),$$

(31)

where

$$k_1(\lambda_n, s) = \left(\lambda_n^2 + \frac{F}{J_3}\right) s^2 + \lambda_n^4, \quad F_{q+1,k}(\lambda_n) = 2\lambda_n \left(\Lambda_{11}^{(q)} \delta_{1k} - \Lambda_{11}^{(q)} \lambda_n^2 \delta_{2k} + D_1^{(q)} \delta_{q+2,k} - \Lambda_{11}^{(q)} \sum_{j=1}^{N} \alpha_1^{(j)} \delta_{j+2,k}\right),$$

$$k_{q+1}(\lambda_n, s) = s + \tau_q s^2 + D_1^{(q)} \lambda_n^2, \quad F_{1k}(\lambda_n) = -2\lambda_n \delta_{1k} + 2\lambda_n \left(s^2 + \lambda_n^2\right) \delta_{2k},$$

$$G_{mk}^L(x,s) = \sum_{n=1}^{\infty} G_{mk}^{Ls}(\lambda_n, s) \sin \lambda_n x, \quad G_{mk}^{Ls}(\lambda_n, s) = 2 \int_0^1 G_{mk}^L(x,s) \sin \lambda_n x \, dx, \quad \lambda_n = \pi n.$$

The solution of system (31) has the following form:

$$G_{1k}^{Ls}(\lambda_n, s) = \frac{P_{1k}(\lambda_n, s)}{P(\lambda_n, s)}, \quad G_{q+1,1}^{Ls}(\lambda_n, s) = \frac{2\lambda_n \Lambda_{11}^{(q)}}{k_{q+1}(\lambda_n, s)} + \frac{P_{q+1,1}(\lambda_n, s)}{Q_q(\lambda_n, s)},$$

$$G_{q+1,2}^{Ls}(\lambda_n, s) = -\frac{2\Lambda_{11}^{(q)} \lambda_n^3}{k_{q+1}(\lambda_n, s)} + \frac{P_{q+1,2}(\lambda_n, s)}{Q_q(\lambda_n, s)},$$

(32)

$$G_{q+1,p+2}^{Ls}(\lambda_n, s) = \frac{2\lambda_n \left(D_1^{(q)} \delta_{qp} - \Lambda_{11}^{(q)} \alpha_p\right)}{k_{q+1}(\lambda_n, s)} + \frac{P_{q+1,p+2}(\lambda_n, s)}{Q_q(\lambda_n, s)} \quad (q, p = \overline{1, N}, \quad k = \overline{1, N+1}).$$

Here,

$$P(\lambda_n, s) = k_1(\lambda_n, s)\Pi(\lambda_n, s) - \lambda_n^6 \sum_{j=1}^{N} \Lambda_{11}^{(j)} \alpha_1^{(j)} \Pi_j(\lambda_n, s), \quad Q_q(\lambda_n, s) = k_{q+1}(\lambda_n, s)P(\lambda_n, s),$$

$$P_{11}(\lambda_n, s) = -2\lambda_n \left[ \Pi(\lambda_n, s) - \lambda_n^2 \sum_{j=1}^{N} \alpha_1^{(j)} \Lambda_{11}^{(j)} \Pi_j(\lambda_n, s) \right],$$

$$P_{12}(\lambda_n, s) = 2\lambda_n \left[ \left(s^2 + \lambda_n^2\right)\Pi(\lambda_n, s) - \lambda_n^4 \sum_{j=1}^{N} \alpha_1^{(j)} \Lambda_{11}^{(j)} \Pi_j(\lambda_n, s) \right],$$

$$P_{1,q+2}(\lambda_n, s) = 2\alpha_1^{(q)}\lambda_n^3 \left[ D_1^{(q)} \Pi_q(\lambda_n, s) - \sum_{j=1}^{N} \alpha_1^{(j)} \Lambda_{11}^{(j)} \Pi_j(\lambda_n, s) \right],$$

$$P_{q+1,1}(\lambda_n, s) = 2\lambda_n^5 \Lambda_{11}^{(q)} \left[ \Pi(\lambda_n, s) - \lambda_n^2 \sum_{j=1}^{N} \alpha_1^{(j)} \Lambda_{11}^{(j)} \Pi_j(\lambda_n, s) \right],$$

$$P_{q+1,2}(\lambda_n, s) = 2\lambda_n^5 \Lambda_{11}^{(q)} \left[ \left(s^2 + \lambda_n^2\right)\Pi(\lambda_n, s) - \lambda_n^4 \sum_{j=1}^{N} \alpha_1^{(j)} \Lambda_{11}^{(j)} \Pi_j(\lambda_n, s) \right],$$

$$P_{q+1,p+2}(\lambda_n, s) = 2\alpha_1^{(p)}\lambda_n^7 \Lambda_{11}^{(q)} \left[ D_1^{(p)} \Pi_p(\lambda_n, s) - \sum_{j=1}^{N} \alpha_1^{(j)} \Lambda_{11}^{(j)} \Pi_j(\lambda_n, s) \right],$$

$$\Pi(\lambda_n, s) = \prod_{j=1}^{N} k_{j+1}(\lambda_n, s), \quad \Pi_j(\lambda_n, s) = \prod_{r=1, r \neq j}^{N} k_{r+1}(\lambda_n, s).$$

The Laplace originals of functions in (32) have the following form [28]:

$$G_{1k}^s(\lambda_n, \tau) = \sum_{j=1}^{2N+2} A_{1k}^{(j)}(\lambda_n)e^{s_j(\lambda_n)\tau}, \quad \xi_{1,2}(\lambda_n) = \frac{-1 \pm \sqrt{1 - 4\tau_q D_1^{(q)}\lambda_n^2}}{2\tau_q},$$

$$G_{q+1,1}^s(\lambda_n, \tau) = \sum_{l=1}^{2} \left[ \frac{2\Lambda_{11}^{(q)}\lambda_n}{1 + 2\tau_q\xi_l(\lambda_n)} + A_{q+1,1}^{(2N+2+l)}(\lambda_n) \right] e^{\xi_l(\lambda_n)\tau} + \sum_{j=1}^{2N+2} A_{q+1,1}^{(j)}(\lambda_n)e^{s_j(\lambda_n)\tau},$$

$$G_{q+1,2}^s(\lambda_n, \tau) = \sum_{l=1}^{2} \left[ \frac{-2\Lambda_{11}^{(q)}\lambda_n^3}{1 + 2\tau_q\xi_l(\lambda_n)} + A_{q+1,2}^{(2N+2+l)}(\lambda_n) \right] e^{\xi_l(\lambda_n)\tau} + \sum_{j=1}^{2N+2} A_{q+1,2}^{(j)}(\lambda_n)e^{s_j(\lambda_n)\tau},$$

$$G_{q+1,p+2}^s(\lambda_n, \tau) = \sum_{l=1}^{2} \left[ \frac{2\left(D_1^{(q)}\delta_{qp} - \Lambda_{11}^{(q)}\alpha_1^{(p)}\right)\lambda_n}{1 + 2\tau_q\xi_l(\lambda_n)} + A_{q+1,p+2}^{(2N+2+l)}(\lambda_n) \right] e^{\xi_l(\lambda_n)\tau} + \sum_{j=1}^{2N+2} A_{q+1,p+2}^{(j)}(\lambda_n)e^{s_j(\lambda_n)\tau},$$

$$A_{1k}^{(j)}(\lambda_n) = \frac{P_{1k}\left(\lambda_n, s_j(\lambda_n)\right)}{P'\left(\lambda_n, s_j(\lambda_n)\right)} \quad (j = \overline{1, N+3}, \quad k = \overline{1, N+2}, \quad r = \overline{1, N+4}, \quad q, p = \overline{1, N}),$$

$$A_{q+1,k}^{(r)}(\lambda_n) = \frac{P_{q+1,k}\left(\lambda_n, s_r(\lambda_n)\right)}{Q'_q\left(\lambda_n, s_r(\lambda_n)\right)}, \quad A_{q+1,k}^{(2N+2+l)} = \frac{P_{q+1,k}\left(\lambda_n, \xi_l(\lambda_n)\right)}{Q'_q\left(\lambda_n, \xi_l(\lambda_n)\right)} \quad (l = 1, 2),$$

where $s_j(\lambda_n)$, $j = \overline{1, 2N+2}$ are zeros of polynomial $P(\lambda_n, s)$.

## 4. Examples

For the calculation example, we take the one-component ($N = 1$) aluminum medium with the following characteristics [31]:

$$C_{1111}^* = 12.05 \cdot 10^{10} \frac{N}{m^2}, \quad T_0 = 800\,K, \quad \rho = 2700 \frac{kg}{m^3},$$

$$\alpha_{11}^{*(1)} = 4.2 \cdot 10^5 \frac{J}{mol}, \quad D_{11}^{*(1)} = 7.73 \cdot 10^{-14} \frac{m^2}{s}, \quad L = 0.1\,m, \quad \tau^{(q)} = 200\,s.$$

We assume that the beam has a rectangular section with height $h = 0.1\,m$ and width $b = 0.05\,m$. The geometrical characteristics of the section is as follows:

$$F = bh = 5.00 \cdot 10^{-5}\,m^2, \quad J_3 = \frac{bh^3}{12} = 4.17 \cdot 10^{-10}\,m^4.$$

*Example 1.* We assume in the boundary conditions (19), (24), and (26):

$$f_{11}(\tau) = -\frac{M_0(\tau)}{J_3} = H(\tau), \quad f_{12}(\tau) = -\frac{M_1(\tau)}{J_3} = H(\tau),$$

$$f_{21}(\tau) = V_0(\tau) = 0, \quad f_{22}(\tau) = V_1(\tau) = 0, \quad f_{31}(\tau) = H_{20}(\tau) = 0, \quad f_{32}(\tau) = H_{21}(\tau) = 0.$$

Then, according to (28), in the absence case of volume perturbations, we have the following:

$$v(x,\tau) = \int_0^\tau [G_{11}(x,\tau-t) + G_{11}(1-x,\tau-t)]H(t)dt =$$

$$= 2\sum_{n=1}^\infty \sin\frac{\lambda_n}{2}\cos\lambda_n\left(\frac{1}{2}-x\right)\sum_{j=1}^4 A_{11}^{(j)}(\lambda_n)\frac{e^{s_j(\lambda_n)\tau}-1}{s_j(\lambda_n)},$$

$$H_1(x,\tau) = \int_0^\tau [G_{21}(x,\tau-t) + G_{21}(1-x,\tau-t)]H(t)dt = \qquad (33)$$

$$= 2\sum_{n=1}^\infty \sin\frac{\lambda_n}{2}\cos\lambda_n\left(\frac{1}{2}-x\right)\sum_{l=1}^2\left(\frac{2\Lambda_{11}^{(1)}\lambda_n}{1+2\tau_q\xi_l(\lambda_n)} + A_{21}^{(4+l)}(\lambda_n)\right)\frac{e^{\xi_l(\lambda_n)\tau}-1}{\xi_l(\lambda_n)} +$$

$$+ 2\sum_{n=1}^\infty \sin\frac{\lambda_n}{2}\cos\lambda_n\left(\frac{1}{2}-x\right)\sum_{j=1}^4 A_{21}^{(j)}(\lambda_n)\frac{e^{s_j(\lambda_n)\tau}-1}{s_j(\lambda_n)}.$$

The calculation results are shown in Figure 2. It should be noted that the effects associated with the diffusion flux relaxation for given perturbations (33) do not appear at all. The graphs for the beam deflection $v(x,\tau)$ and the concentration increment $H_1(x,\tau)$ without considering the relaxation have the same view as in Figure 2.

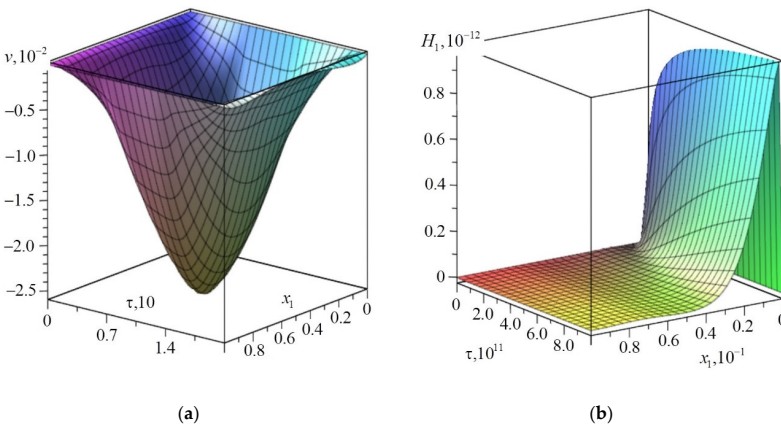

(a)                                                                                (b)

**Figure 2.** The results of calculations using Equation (34): (**a**) The beam deflections $v(x,\tau)$; (**b**) The concentration increment $H_1(x,\tau)$.

*Example 2.* Now, we assume in the boundary conditions (19), (24), and (26) that

$$f_{11}(\tau) = -\frac{M_0(\tau)}{J_3} = 0, \quad f_{12}(\tau) = -\frac{M_1(\tau)}{J_3} = 0,$$

$$f_{21}(\tau) = V_0(\tau) = 0, \quad f_{22}(\tau) = V_1(\tau) = 0, \quad f_{31}(\tau) = H_{20}(\tau) = H(\tau), \quad f_{32}(\tau) = H_{21}(\tau) = H(\tau).$$

That (28)

$$v(x,\tau) = \int_0^{\tau} [G_{13}(x,\tau-t) + G_{13}(1-x,\tau-t)]H(t)dt =$$

$$= 2\sum_{n=1}^{\infty} \sin\frac{\lambda_n}{2}\cos\lambda_n\left(\frac{1}{2}-x\right)\sum_{j=1}^{4} A_{13}^{(j)}(\lambda_n)\frac{e^{s_j(\lambda_n)\tau}-1}{s_j(\lambda_n)},$$

$$H_1(x,\tau) = \int_0^{\tau} [G_{22}(x,\tau-t) + G_{22}(1-x,\tau-t)]H(t)dt =$$

$$= 2\sum_{n=1}^{\infty} \sin\frac{\lambda_n}{2}\cos\lambda_n\left(\frac{1}{2}-x\right)\sum_{l=1}^{2}\left(\frac{2\left(D_1^{(1)}-\alpha_1^{(1)}\Lambda_{11}^{(1)}\right)\lambda_n}{1+2\tau_q\xi_l(\lambda_n)} + A_{23}^{(4+l)}(\lambda_n)\right)\frac{e^{\xi_l(\lambda_n)\tau}-1}{\xi_l(\lambda_n)}+$$

$$+2\sum_{n=1}^{\infty} \sin\frac{\lambda_n}{2}\cos\lambda_n\left(\frac{1}{2}-x\right)\sum_{j=1}^{4} A_{23}^{(j)}(\lambda_n)\frac{e^{s_j(\lambda_n)\tau}-1}{s_j(\lambda_n)}.$$

The calculation results are shown in Figure 3.

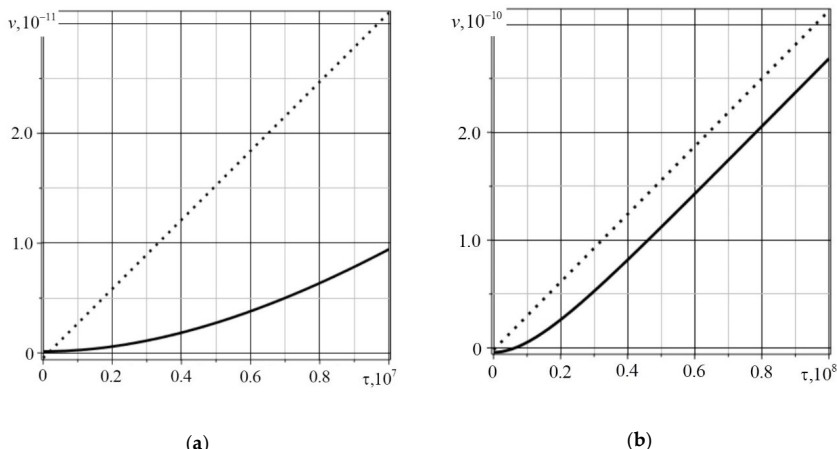

(a)  (b)

**Figure 3.** The beam deflections $v(x,\tau)$, $x = 0.5$. The solid line is the solution with relaxation of diffusion fluxes, and the dashed line is the solution without relaxation of diffusion fluxes: (**a**) Effect of diffusion processes relaxation on the beam deflections at $\tau \sim 10^7$; (**b**) Effect of diffusion processes relaxation on the beam deflections at $\tau \sim 10^8$.

Relaxation effects are manifested under the condition $\tau \leq \tau_q$ (dimensionless time $10^7$ is corresponds to 150 s), as can be seen in Figure 3. The relaxation effects fade out with increasing time, which is demonstrated in Figure 4. A similar situation is observed for the concentration increment $H_q(x,\tau)$. Based on the completed study, it can be concluded that the relaxation effects manifestation essentially depends on both the material properties and the type of specified perturbations. The results obtained in this article are consistent with the results obtained by other researchers, for example, [6,12,32,33].

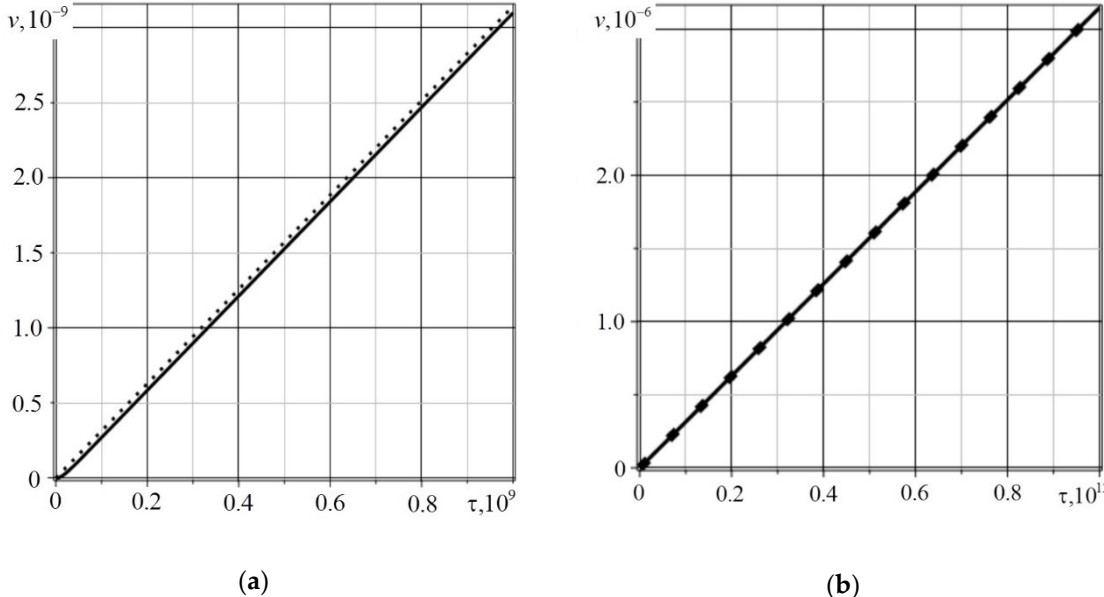

**Figure 4.** The beam deflections $v(x, \tau)$, $x = 0.5$. The solid line is the solution with relaxation of diffusion fluxes, and the dashed line is the solution without relaxation of diffusion fluxes: (**a**) Effect of diffusion processes relaxation on the beam deflections at $\tau \sim 10^9$; (**b**) Effect of diffusion processes relaxation on the beam deflections at $\tau \sim 10^{12}$.

## 5. Conclusions

The coupled unsteady oscillations model of an elastic diffusion Euler–Bernoulli beam was constructed using Hamilton's variational principle. An algorithm for constructing surface Green's functions was proposed. The use of the unknown function expansion into series by eigenfunctions allowed us to solve the problem associated with the Laplace transform inversion. Such an approach made it possible to find an analytical solution to the oscillation problem of an elastic diffusion Euler–Bernoulli beam.

On the basis of the developed model, the interaction of mechanical and diffusion fields was investigated. The influence of diffusion relaxation effects on the beam was analyzed. A number of test calculations showed that relaxation effects appeared only at the initial moments of time and under certain boundary conditions. The results of the calculations are presented in analytical and graphical forms.

**Author Contributions:** D.T. and A.Z. contributed equally to the writing of the main manuscript and preparation of the figures. Both authors reviewed the manuscript. Both authors contributed equally to the paper.

**Funding:** This work was funded by a subsidy from RFBR (Project 17-08-00663 A).

**Conflicts of Interest:** The funders had no role in the design of the study; in the collection, analyses, or interpretation of data; in the writing of the manuscript; or in the decision to publish the results.

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
