# Peer review of "An Elastodiffusive Orthotropic Euler–Bernoulli Beam Considering Diffusion Flux Relaxation"

_mca, doi:10.3390/mca24010023_

Round 1

Reviewer 1 Report

The title of the article is fully consistent with its content. The article has a clear structure, which includes all the main sections: problem analysis, formulation and solution of the problem, analysis of the obtained results and the references list. The calculations results are presented in analytical form, as well as in the graph form. Thus, content and information of the article are presented clearly and understandably.

A significant advantage of the presented article is the development of the analytical method for solving the unsteady elastic diffusion problem of an Euler – Bernoulli beam vibrations. This is important, since such problems are solved mainly in static and quasistatic formulations. Attempts are known to use numerical algorithms for solving unsteady mechanodiffusion problems, for example, finite-difference schemes. Here the main issue is the finite-difference schemes stability analysis. This is quite a complex mathematical problem. At the same time, analytical methods allow us to obtain the exact solution of the problem in an explicit form.

The disadvantages of the work include the following:

1) The evaluation of the effect of mass transfer on the stress-strain state of the beam was not performed. In addition, it is known that diffusion is a very slow process. Therefore, the time during which mass transfer may affect the bearing capacity of structures may be greater than the standard lifetime of the structure.

2) The assessment of the effect of anisotropy on the physical fields inside the beam was not carried out.

3) There are typos. Judging by the formulas in line 154, all three-dimensional Green's functions must contain in their designation the “~” sign. Corresponding amendments should be made in lines 145 and 153.

However, these comments do not reduce the scientific value of the work. The article may be published after correcting the typos.

Author Response

Authors thank Reviewer for the attentively article reading. Regarding the comments, we report the following:

1. We agree with the remark and took it into account.

2. We agree with the remark. The issue of the anisotropy effect on mass transfer is not considered in the article.

3. Corrections made.

Reviewer 2 Report

1. Although the Introduction provides numerous literature sources, the analysis of those sources is rather insufficient. An introduction is to clearly present the state-of-the-art of this particular problem, characterize current challenges and gaps in the relevant knowledge, and, what is most important, to clarify how the presented research fills this gap. The Authors are advised to revise the Introduction accordingly.

2. The Conclusion is rather insufficient and seems to be even shorter than the Abstract.

3. In Fig. 1, the cross-section of the considered beam is denoted with $D$; the boundary is denoted with $\Gamma$. In what follows (line 60 on page 3), the domain of consideration is $G$ with boundary $\partial G$. What is the connection between these objects?

4. It is assumed (by “i" in line 69 on page 3) that the Cartesian axis  $x_3$ is the central axis of the cross-section. Does this imply that the beam is a body of revolution?

5. The English needs a thorough revision and improvement by a native speaker, preferably. The vast majority of statements seem to be translated with the direct implication of Russian grammar, which is hard to follow. The verbs are missing in some sentences (for example, the second sentence of Introduction), etc.

Author Response

Authors thank Reviewer for the attentively article reading. Regarding the comments, we report the following:

1.    Introduction corrected.

2.    Conclusion corrected.

3.    $D \subset \Re^2$ is cross-section area, $G \subset \Re^3$ is the beam material space. Accordingly, $\Gamma$ is the boundary of $D$ and $\partual G$ is the boundary of $G$ (beam surface).

4.    The axis $Ox_3$ passes through the mass center of the beam cross-section. It means that the static moment of inertia around this axis is zero (formula 1.7)

5.    Unfortunately, we are not able to personally contact the native speakers with a request for verification. We once again independently checked and corrected the translation of the article into English. The final decision on this issue is left to the discretion of the journal editors.
